# Immune-Modified Glasgow Prognostic Score Predicts Therapeutic Effect of Pembrolizumab in Recurrent and Metastatic Head and Neck Cancer

**DOI:** 10.3390/cancers16234056

**Published:** 2024-12-03

**Authors:** Natsuko Ueda, Masashi Kuroki, Hirofumi Shibata, Manato Matsubara, Saki Akita, Tatsuhiko Yamada, Rina Kato, Ryota Iinuma, Ryo Kawaura, Hiroshi Okuda, Kosuke Terazawa, Kenichi Mori, Ken Saijo, Toshimitsu Ohashi, Takenori Ogawa

**Affiliations:** 1Department of Otolaryngology-Head and Neck Surgery, Graduate School of Medicine, Gifu University, Gifu 501-1194, Japan; obaue926@yahoo.co.jp (N.U.); kurokim34@gmail.com (M.K.); manakame1004@gmail.com (M.M.); coco69927@gmail.com (S.A.); ta2hiko.yama@gmail.com (T.Y.); rina_1699@yahoo.co.jp (R.K.); ryota.i890302@gmail.com (R.I.); m2001023@gmail.com (R.K.); h_okuda_gifu@yahoo.co.jp (H.O.); course._.k@me.com (K.T.); t2111030@gmail.com (K.M.); o_1043_toshi32_dragons@yahoo.co.jp (T.O.); 2Department of Clinical Oncology, Graduate School of Medicine, Tohoku University, Sendai 980-8576, Japan; ken.saijo.d6@tohoku.ac.jp; 3Department of Head and Neck Surgery-Otorhinolaryngology, Ogaki Municipal Hospital, Ogaki 503-8502, Japan

**Keywords:** head and neck cancer, ALC, NLR, GPS, mGPS

## Abstract

In recent years, pembrolizumab has become applicable for recurrent and metastatic head and neck cancer (RMHNC), but only a limited number of patients can benefit from it. In this study, we explored predictive markers for the therapeutic effect and prognosis of pembrolizumab for RMHNC. The results showed that the immune-modified Glasgow Prognostic Score (imGPS), which we previously proposed, is a better predictive marker than the previously known neutrophil-to-lymphocyte ratio (NLR) and mGPS. The imGPS can be easily evaluated with just a blood test, so we hope that it will become more widely used in the clinical environment.

## 1. Introduction

Traditionally, the treatment of recurrent and metastatic head and neck cancer (RMHNC) has relied on drug therapy, primarily cytotoxic agents such as platinum-based drugs and the molecular-targeted drug cetuximab [1]. Recently, immunotherapy for head and neck cancer has been developed, and results of the KEYNOTE-040 and KEYNOTE-048 trials have confirmed the efficacy of the immune checkpoint inhibitor (ICI) pembrolizumab for RMHNC [2,3]. The Combined Positive Score (CPS), used as a biomarker in the KEYNOTE-048 trial, is now widely used in clinical practice as a complementary diagnostic method [3]. Tumor Mutation Burden (TMB) represents the mutation burden per megabase and is considered a biomarker to predict the therapeutic effect of the ICI [4,5,6]. The TMB measurement incurs considerable expense for next-generation sequencing. Therefore, TMB is often measured after an ICI regimen is administered and is rarely used as a pretreatment indicator in clinical practice [7].

Alternatively, various biomarkers, easily calculated from blood data or nutritional status, have been reported. The neutrophil-to-lymphocyte ratio (NLR) is calculated using peripheral blood and serves as a predictive marker for prognosis and therapeutic effect in various cancers [8,9]. The Glasgow Prognostic Score (GPS), first proposed by Forrest et al. (2003) and reported by the same group under the name “GPS” in 2004 [10,11], is a cumulative prognostic score based on systemic inflammatory response and albumin levels. Thereafter, based on the finding that ALB alone does not correlate with prognosis, a new term, the “modified GPS (mGPS)”, was proposed by McMillan et al. (2007) [12,13]. Many reports have shown that the GPS/mGPS is an independent prognostic marker in various cancers [14]. Because acquired immunity influences cancer progression and recurrence, our group proposed the immune-modified GPS (imGPS), which adds the peripheral blood lymphocyte count as a component of the mGPS [15]. Our study analyzed 461 head and neck squamous cell carcinoma (HNSCC) cases and demonstrated that imGPSs can be a more robust prognostic marker than mGPSs [15].

In this study, using imGPSs to unveil the efficacy of imGPSs in ICI-treated patients, we analyzed the prognosis and therapeutic effect of pembrolizumab for RMHNC patients.

## 2. Materials and Methods

### 2.1. Patients

This study included 54 patients with RMHNC who were treated with pembrolizumab at Gifu University Hospital (Japan) from December 2019 to April 2024. Pembrolizumab treatment was used for RMHNC patients who had no history of chemotherapy for the target disease. Basically, patients were divided into pembrolizumab-alone and chemotherapy combination groups based on the CPS, but the regimen was selected taking into consideration not only the CPS but also age, Performance Status (PS), and the tumor progression rate. In the chemotherapy combination group, cisplatin + 5-fluorouracil or carboplatin + 5-fluorouracil was selected based on renal function, and some cases included a dose reduction. All relevant clinical data were acquired from patient medical records. This study was approved by the Institutional Ethical Committee of the Gifu University Graduate School of Medicine (No.2023-253) and was conducted in accordance with the Helsinki Declaration.

### 2.2. Data Collection and Endpoints

CRP, ALB, and lymphocyte and neutrophil counts were extracted from blood test data collected immediately before the administration of pembrolizumab, from which the NLR, mGPS, and imGPS were calculated. CRP, ALB, and lymphocyte counts were determined using cutoff values of the imGPS, which have been previously reported (CRP = 1 mg/dL, ALB = 3.5 g/dL, and lymphocyte count = 1250/μL) [15]. The NLR was calculated as the neutrophil/lymphocyte count, and the cutoff value was determined using a receiver operating characteristic (ROC) analysis. Scoring methods for the mGPS and imGPS are shown in Table 1. In short, the mGPS was based on CRP and ALB, whereas the imGPS was based on CRP and ALB values plus lymphocyte counts. The CPS was calculated as follows: existing tissue specimens obtained by biopsy or surgery were immunostained with the anti-PD-L1 mouse monoclonal antibody (Clone22C3), and the number of positive cells (tumor cells, lymphocytes, and macrophages) was divided by the total number of viable tumor cells and multiplied by 100. Treatment effects were evaluated as the best overall response (BOR), and the overall response rate (ORR) was calculated as the proportion of PR (partial response) and CR (complete response) cases among the total number of cases. The prognosis was evaluated as overall survival (OS) and progression-free survival (PFS). Each predictive marker was analyzed separately for the pembrolizumab-alone and the chemotherapy combination groups.

### 2.3. Statistical Analysis

Survival time analysis was performed using the Kaplan–Meier method, and differences between groups were analyzed using the Log-rank test. Differences in the ORR between groups were analyzed using the Chi-square test. The significance of associated parameters was assessed using the Cox proportional hazards regression model. To evaluate the utility of the imGPS for the predictive marker of prognosis (OS and PFS), ROC analysis was performed to calculate the area under the curve (AUC) value. All statistical analyses were performed using EZR (version 3.6.3). In all analyses, a *p*-value < 0.05 was considered significant.

## 3. Results

### 3.1. Patient Characteristics

A total of 54 patients were enrolled (Table 2). The median age was 70 years (range: 37–91). The primary site was diverse with the most common being the oropharynx at 27.8%. Pathologically, squamous cell carcinoma (SCC) accounted for 83.3%. Cases of the CPS < 1 had a frequency of 9.3%. Cases with 1 ≤ CPS < 20 totaled 55.6%, and cases of 20 ≤ CPS amounted to 27.8%. Regimens included pembrolizumab alone in 48.1% of these cases and in combination with therapeutants, CDDP and 5-FU, in 51.9% of all cases. The median follow-up period was 10.6 months, ranging from 0.1 to 45.2 months.

The NLR, mGPS, and imGPS were then calculated from the blood test data. The cutoff value for the NLR was set at 6.589 based on the ROC curve (AUC: 0.585, 95% CI: 0.43–0.739) (Appendix A). For the mGPS, 37 cases scored 0, 5 cases scored 1, and 12 cases scored 2, whereas for the imGPS, 14 cases scored 0, 21 cases scored 1, 9 cases scored 2, and 10 cases scored 3, indicating that the imGPS was able to allocate cases more evenly.

### 3.2. A Predictive Marker for the Therapeutic Effect of Pembrolizumab

After pembrolizumab administration, a complete response (CR) was observed in 7 cases (13.0%), a partial response (PR) in 8 cases (14.8%), stable disease (SD) in 12 cases (22.2%), and progressive disease (PD) in 21 cases (38.9%, with the ORR being 31.3% (Appendix A)). No significant differences were observed in individual factors between the pembrolizumab-alone and the chemotherapy combination groups.

First, to show the utility of the imGPS, correlations between its components (CRP, ALB, and lymphocyte count) and the therapeutic effect of pembrolizumab were investigated. The group with lymphocyte counts ≥1250/μL had a significantly improved ORR compared with the group with lymphocyte counts <1250/μL (<1250/μL: 18.6%, ≥1250/μL: 56.3%, *p* = 0.0082) (Figure 1). However, no significant differences in treatment efficacy were observed for CPR and ALB. A multivariate analysis was performed with age, chemotherapy, and the imGPS components as confounding factors, but only the lymphocyte count showed a significant difference (Table 3).

Next, in order to investigate the utility of the imGPS as a predictive marker for therapeutic effect, the previously reported NLRs and mGPSs were also examined for a correlation with the therapeutic effect of pembrolizumab. No significant differences in treatment efficacy were observed for the NLR or mGPS. Only the imGPS showed a significant difference (score 0: 57.1%, score 1: 22.2%, score 2: 12.5%, and score 3: 25%) (Figure 1). These data suggest that the imGPS is a robust tool to predict the response to pembrolizumab treatment for RMHNC.

### 3.3. A Predictive Marker for Prognosis of Pembrolizumab

The prognosis for all cases had a median OS of 14.5 months and a median PFS of 3.4 months (Figure 2). When examining OS/PFS by age, CPS status, and whether chemotherapy was administered, the differences were not significant (Appendix A).

First, we examined the correlation between the imGPS components (CRP, ALB and lymphocyte count) and the prognosis of pembrolizumab (Figure 3). The low-CRP group (<1 mg/dL) had significantly improved OS and PFS (median OS: 17.6 months vs. 7.9 months: *p* = 0.0023, median PFS: 6.0 months vs. 1.2 months: *p* = 0.0013). The high-ALB group (≥3.5 g/dL) had significantly improved OS (median OS: 7.9 months vs. 17.6 months: *p* = 0.0016), and the high-lymphocyte-count group (≥1250/μL) showed significantly improved OS and PFS (median OS: 11.2 months vs. NA: *p* = 0.0008, median PFS: 2.4 months vs. 11.0 months: *p* = 0.0014). A multivariate analysis was performed with age, chemotherapy, and imGPS components as confounding factors. Significant differences were observed in CRP and lymphocyte count for OS and lymphocyte count for PFS (Table 4). We examined prognosis separately for the pembrolizumab-alone and chemotherapy combination groups. The results for the pembrolizumab-alone group were similar to those in the all-cases group (Appendix A).

Next, we examined the correlation between each predictive marker (the NLR, mGPS, and imGPS) and prognosis (Figure 4). There was no significant difference in OS with the NLR, but the group with a lower score (6.589 or less) had a better PFS (median OS: 14.9 months vs. 5.8 months: *p* = 0.0992, median PFS: 6.0 months vs. 1.6 months: *p* = 0.0295). The group with a lower mGPS had a better OS and PFS (median OS: 17.6 months vs. 14.3 months vs. 6.8: *p* = 0.0034, median PFS: 6.0 months vs. 1.6 months vs. 2.3 months: *p* = 0.0127). Similarly, the group with a lower imGPS had a better OS and PFS (median OS: NA vs. 12.0 months vs. 11.8 months vs. 6.1: *p* = 0.0011, median PFS: 11.0 months vs. 3.9 months vs. 2.1 months vs. 2.2 months: *p* = 0.0049). Furthermore, we examined the prognosis separately for the pembrolizumab-alone and chemotherapy-combination groups. The results for the pembrolizumab-alone group were similar to those in the all-cases group. ROC analysis was performed to compare the utility of the imGPS with the NLR and mGPS (Table 5). Intriguingly, the imGPS showed the highest AUC values for both OS and PFS, suggesting that adding peripheral lymphocyte counts enhances the prediction of pembrolizumab-reactive RMHNC patients (Appendix A). These data highlight the imGPS’s superiority over other predictive markers or scales.

## 4. Discussion

In this study, we showed that CRP, ALB, lymphocyte counts, NLRs, mGPSs, and imGPSs were correlated with prognosis, and that the imGPS can be a robust predictive marker for pembrolizumab efficacy in RMHNC compared with the NLR and mGPS. Intriguingly, only lymphocyte counts and the imGPS showed statistically significant differences in therapeutic effects. These findings suggest that the imGPS is beneficial not only as a predictive marker for the prognosis of RMHNC but also as a helpful tool for forecasting OS and the therapeutic effect of pembrolizumab in RMHNC. Notably, the imGPS can be calculated using daily peripheral blood data without high-cost tests such as TMB, which require next-generation sequencing. This benefit highlights the potential of imGPSs to identify which RMHNC patients to treat with pembrolizumab.

The reason that the imGPS is more useful than the mGPS is that lymphocytes in peripheral blood are important in immunotherapy, and there are reports that peripheral blood lymphocytes correlate with prognosis [16,17,18,19,20]. Many reports stating that peripheral blood lymphocytes are correlated with prognosis do not mention the relationship between the tumor microenvironment and peripheral blood lymphocytes. Wu et al. performed the single-cell sequencing of RNA and T cell receptors in patients with different types of cancer. They found that the clonotypic expansion of effector-like T cells in tumor tissue also occurred in peripheral blood [16]. In our study, an increase in peripheral blood lymphocyte count was also associated with therapeutic efficacy, suggesting that the peripheral blood lymphocyte count may reflect the immune response of the tumor microenvironment during ICI treatment. Moreover, some reports state that there is a correlation between Tumor Infiltrating Lymphocytes (TILs) and peripheral blood lymphocytes [20]. Therefore, it is expected that peripheral blood lymphocytes are also related to the therapeutic mechanism of ICIs. Patients in this study treated with pembrolizumab did not have a history of being treated with chemotherapy for the disease, and these results appeared to directly reflect the efficacy of ICI treatment without disturbing the acquired immune state. In this study, although the lymphocyte count alone was able to predict prognosis, no significant differences were found in the NLR, which included the lymphocyte count. This may seem contradictory at first glance. HNCs develop close to the outside with frequent contact with microorganisms, which induces inflammation. In addition, some advanced HNCs tend to show necrosis because of a lack of blood flow in the center region of the tumor. These factors influence neutrophil counts and may be related to the weak effectiveness of the NLR compared to the imGPS in correctly predicting pembrolizumab efficacy.

In several studies, the GPS/mGPS has proven to be a predictive marker for prognosis in RMHNC treated with nivolumab [21,22,23,24,25]. All reports found that the GPS/mGPS was correlated with prognosis, but none of the reports analyzed it as a predictive marker for therapeutic effect. Although there have been many similar reports regarding nivolumab, few reports have examined the prognostic and therapeutic value of pembrolizumab in RMHNC using blood test data alone, as in our study. Previous reports on predictive markers for the prognosis and therapeutic effect of pembrolizumab in HNC are summarized in Table 6 [26,27,28,29,30,31]. A past study showed no correlation between pretreatment CRP and the response rate [29]. Two reports examined lymphocyte counts, but their results conflicted [30,31]. There were seven reports on the NLR, and several of them reported a correlation with prognosis, as in our study, but no correlation with the response rate was found [26,27,28,29,30,31]. Interestingly, other studies have shown that a stronger correlation was observed in the predictive marker during treatment than before treatment, indicating the need for a further analysis of our data [29,31]. We emphasize that this study is the first report to suggest the clinical utility of the imGPS for predicting pembrolizumab efficacy in RMHNC patients.

Limitations of this study are that it was a single-center, retrospective study with a small number of patients. Moreover, it did not consider patient selection biases such as prior treatment history, clinical stage, or histological type. And in this study, patients with imGPSs = 3 had slightly better ORRs and PFSs than those with imGPSs = 2, so there may still be room for an improvement in the scoring method. Further, an accumulation of cases and prospective multi-center studies are needed.

## 5. Conclusions

This study showed that the imGPS can be a robust predictive marker for the prognosis and therapeutic efficacy of pembrolizumab for RMHNC compared with previously reported NLRs and mGPSs. In addition, imGPSs can easily be calculated using only a blood test, making it simple and cost-effective compared to tissue tests. We hope that it will become widely used in clinical practice.

## Figures and Tables

**Figure 1 cancers-16-04056-f001:**
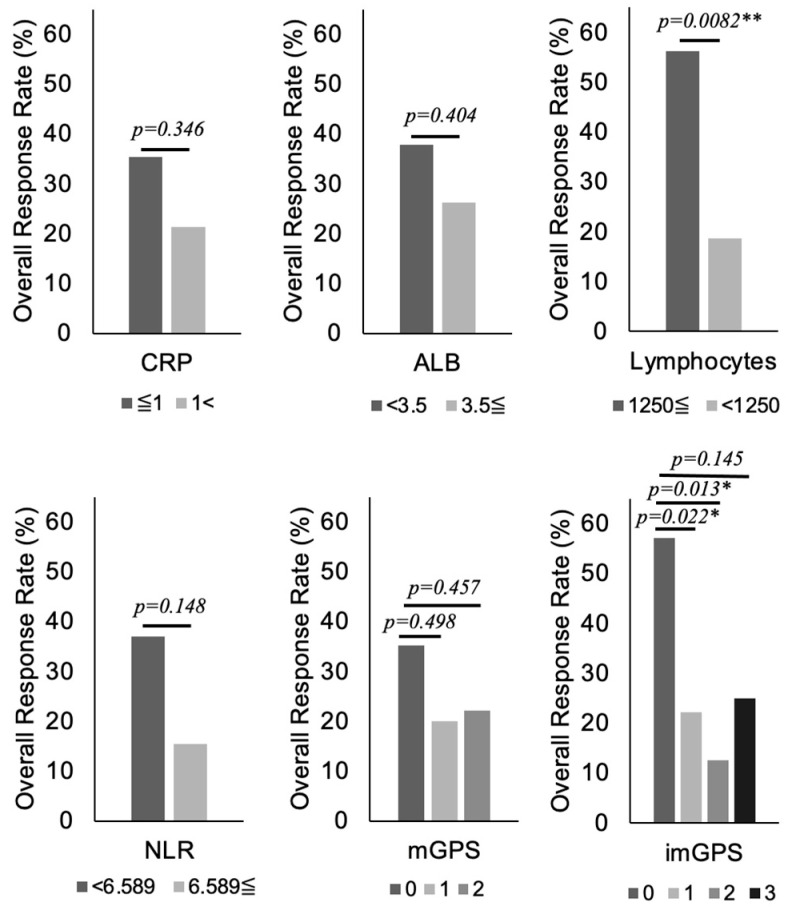
A predictive marker for the therapeutic effect of pembrolizumab. Overall response rates were evaluated for each of the following eight biomarkers: CRP (≦1; *n* = 37, <1; *n* = 17), ALB (<3.5; *n* = 23, ≦3.5; *n* = 31), lymphocytes (≦1250; *n* = 17, <1250; 37), NLR (<6.589; *n* = 38, ≦6.589; *n* = 16), mGPS (0; *n* = 37, 1; *n* = 5, 2; *n* = 12), imGPS (0; *n* = 14, 1; *n* = 21, 2; *n* = 9, 3; *n* = 10). NLR: neutrophil-to-lymphocyte ratio, mGPS: modified Glasgow Prognostic Score, imGPS: immune-modified GPS. **, *p* < 0.01; *, *p* < 0.05.

**Figure 2 cancers-16-04056-f002:**
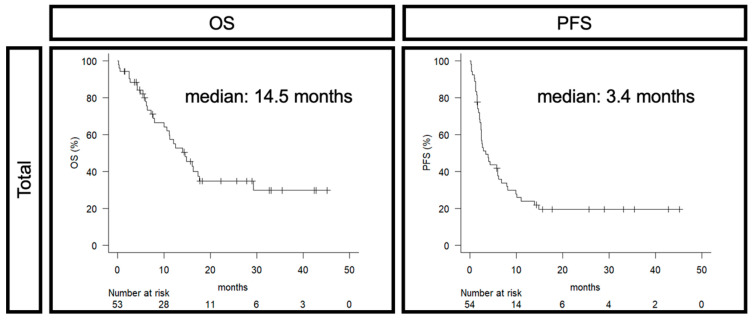
Prognosis of all cases. OS: overall survival, PFS: progression-free survival.

**Figure 3 cancers-16-04056-f003:**
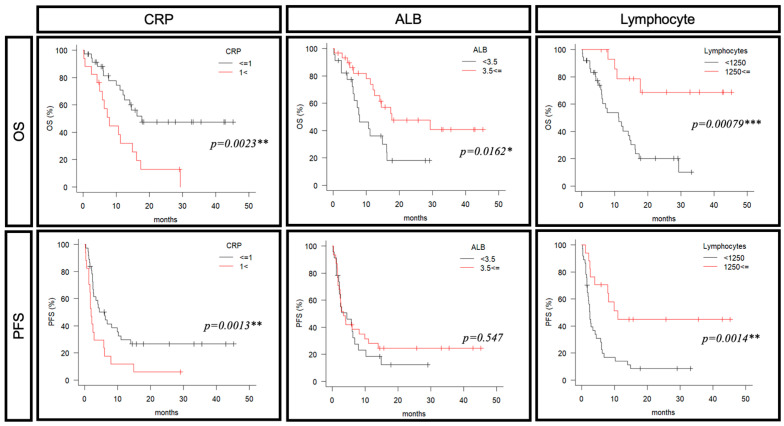
A predictive marker for prognosis of pembrolizumab (CRP, ALB, lymphocytes). OS and progression-free survival (PFS) were evaluated according to CPR and ALB levels and lymphocyte counts; CRP (≦1; *n* = 37, 1<; *n* = 17), ALB (<3.5; *n* = 23, ≦3.5; *n* = 31), lymphocytes (≦1250; *n* = 17, <1250; 37). OS: overall survival, PFS: progression-free survival. ***, *p* < 0.001; **, *p* < 0.01; *, *p* < 0.05.

**Figure 4 cancers-16-04056-f004:**
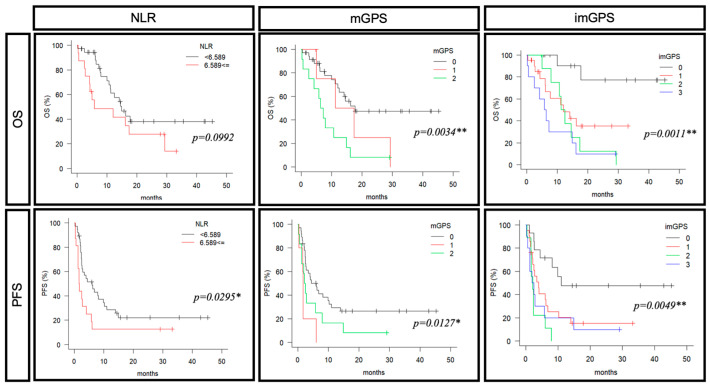
A predictive marker for prognosis of pembrolizumab (NLR, mGPS, imGPS). OS and PFS were evaluated according to NLRs, mGPSs, and imGPSs; NLR (<6.589; *n* = 38, ≦6.589; *n* = 16), mGPS (0; *n* = 37, 1; *n* = 5, 2; *n* = 12), imGPS (0; *n* = 14, 1; *n* = 21, 2; *n* = 9, 3; *n* = 10). OS: overall survival, PFS: progression-free survival, NLR: neutrophil-to-lymphocyte ratio, mGPS: modified Glasgow Prognostic Score, imGPS: immune-modified GPS. **, *p* < 0.01; *, *p* < 0.05.

**Table 1 cancers-16-04056-t001:** Scoring methods for mGPS and imGPS.

mGPS	0		1		2	
CRP	≤1.0		<1			
ALB	-		≦3.5		<3.5	
**imGPS**	**0**	**1**		**2**		**3**
CRP	≤1.0		<1			
ALB	-		≦3.5		<3.5	
Lymphocytes	≦1250	<1250	≦1250	<1250	≦1250	<1250

The conventional mGPS is calculated using only CRP and ALB as indicators, but the imGPS also includes peripheral blood lymphocyte count. mGPS: modified Glasgow Prognostic Score, imGPS: immune-modified GPS.

**Table 2 cancers-16-04056-t002:** Patient characteristics.

Patient Characteristics		*n* = 54	%
**Age**	median (range)	70 (37–91)	
**Sex**	Male	44	81.5
	Female	10	18.5
**Pathology**	SCC	45	83.3
	Other	9	16.7
**Primary**	Nasopharynx	2	3.7
	Oropharynx	9	16.7
	Hypopharynx	15	27.8
	Larynx	5	9.3
	Oral cavity	9	16.7
	Nose/Sinus	10	18.5
	Salivary glands	3	5.6
	Other	1	1.6
**CPS**	<1	5	9.3
	≦1, <20	30	55.6
	≦20	15	27.8
**Regimen**	alone	26	48.1
	combo	28	51.9
**mGPS**	0	37	68.5
	1	5	9.6
	2	12	22.2
**imGPS**	0	14	25.9
	1	21	38.9
	2	9	16.7
	3	10	18.5

SCC: squamous cell carcinoma, CPS: combined positive score, mGPS: modified Glasgow Prognostic Score, imGPS: immune-modified GPS.

**Table 3 cancers-16-04056-t003:** Cox proportional hazards regression analysis for ORR.

		HR	95%CI	*p*-Value
**ORR**	**Age**	0.899	0.1810–4.460	*p* = 0.8960
	**Chemo**	0.722	0.1290–4.060	*p* = 0.7110
	**CRP**	1.000	0.1530–6.550	*p* = 0.9980
	**ALB**	1.600	0.3170–8.080	*p* = 0.5690
	**Lymphocytes**	0.149	0.0297–0.752	*p* = 0.0211 *

ORR: overall response rate, HR: hazard ratio, 95%CI: 95% Confidence Interval. *, *p* < 0.05.

**Table 4 cancers-16-04056-t004:** Cox proportional hazards regression analysis for OS and PFS.

		HR	95%CI	*p*-Value
**OS**	**age**	1.730	0.3840–7.800	*p* = 0.470
	**chemo**	1.180	0.2600–5.330	*p* = 0.8330
	**CRP**	6.670	10.200–44.900	*p* = 0.0478 *
	**ALB**	0.816	0.1930–3.450	*p* = 0.7830
	**Lymphocytes**	4.830	1.1300–20.600	*p* = 0.0331 *
**PFS**	**age**	0.582	0.0930–3.64	*p* = 0.5630
	**chemo**	3.010	0.4640–19.60	*p* = 0.2480
	**CRP**	12.600	0.6410–247.00	*p* = 0.0955
	**ALB**	0.341	0.0485–2.40	*p* = 0.2790
	**Lymphocytes**	9.010	1.5400–52.70	*p* = 0.0146 *

OS: overall survival, PFS: progression-free survival, HR: hazard ratio, 95%CI: 95% Confidence Interval. *, *p* < 0.05.

**Table 5 cancers-16-04056-t005:** Receiver operating characteristic (ROC) analysis for OS and PFS.

		OS		PFS	
		AUC	95%CI	AUC	95%CI
**alone**	**NLR**	0.456	0.173–0.738	0.533	0.182–0.885
	**mGPS**	0.6	0.469–0.731	0.567	0.478–0.656
	**imGPS**	0.678	0.463–0.892	0.65	0.371–0.929
**combo**	**NLR**	0.609	0.393–0.826	0.653	0.381–0.925
	**mGPS**	0.74	0.575–0.904	0.677	0.480–0.873
	**imGPS**	0.833	0.670–0.997	0.786	0.548–1.000
**total**	**NLR**	0.585	0.430–0.739	0.577	0.390–0.765
	**mGPS**	0.703	0.597–0.810	0.644	0.527–0.760
	**imGPS**	0.795	0.682–0.909	0.754	0.599–0.909

ROC: receiver operating characteristic, OS: overall survival, PFS: progression-free survival, NLR: neutrophil-to-lymphocyte ratio, mGPS: modified Glasgow Prognostic Score, imGPS: immune-modified GPS, AUC: area under the curve, 95%CI: 95% Confidence Interval.

**Table 6 cancers-16-04056-t006:** Previous reports investigating predictive markers for prognosis and therapeutic effect of pembrolizumab in head and neck cancer.

Reference No.	Author	Year	*n*	Target	Predictor	Cut-Off Value	Conclusion
[26]	Morimoto, H.	2024	29	R/M HNSCC	NLR	4.5	NLR < 4.5 is associated with better OS and PFS2 No significant difference in PFS
[27]	Lee, R.H.	2024	20	R/M SGC	NLR	5	NLR > 5 is associated with poor PFS and OS
[28]	Sakai, A.	2023	51 (Pembro:15)	R/M HNSCC	NLR	4.5	NLR correlates only with OS No correlation with ORR/DCR or PFS
[29]	Haas, M.	2023	87	R/M HNSCC	CRP	3	No significant difference in response rate (significant difference in CRP during treatment)
					NLR	6	No significant difference in response rate (significant difference in NLR during treatment)
[30]	Ho, W.J.	2023	34 (Pembro:18)	R/M HNSCC	Lymphocytes	600	Lympho < 600 is poor PFS
					NLR	7	NLR ≧ 7 is poor PFS
[31]	Park, J.C.	2020	108 (Pembro:76)	R/M HNSCC	Lymphocytes	700	No significant difference in PFS and ORR (significant difference in lymphocytes during treatment)
					NLR	6.7	No significant difference in PFS and ORR (significant difference in NLR during treatment)

R/M HNSCC: recurrent/metastatic head and neck squamous cell carcinoma, R/M SGC: recurrent/metastatic salivary gland carcinoma.

## Data Availability

The raw data supporting the conclusions of this article will be made available by the authors on request.

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
