# Peer review of "Immune-Modified Glasgow Prognostic Score Predicts Therapeutic Effect of Pembrolizumab in Recurrent and Metastatic Head and Neck Cancer"

_cancers, 2024, doi:10.3390/cancers16234056_

Round 1

Reviewer 1 Report

Comments and Suggestions for Authors

The manuscript titled” Immune-modified Glasgow prognostic score predicts therapeutic effect of pembrolizumab in recurrent and metastatic head and neck cancer” tried to prove the imGPS may be a used predictive and prognostic marker for the therapeutic effect of pembrolizumab for RMHNC. However, there are several defects leading the conclusion might be biased: 1. This is a retrospective study, thus, the power of imGPS on prediction will be limited . 2.  Although 54 patient samples are included, only a small number for each subtype. Besides, 81.5% of included patients are male.

Other comments:

1.      Relative to mGPS, the imGPS added the lymphocyte into consideration. The NLR (Neutrophil-to-Lymphocyte Ratio) was less robust on prediction for prognosis of pembrolizumab than lymphocyte. I wonder if this is due to the variously changed neutrophil levels.

2.      It is great to know that the imGPS that could be employed with daily blood test. I wonder if the imGPS is dynamically changed after receiving treatment if monitored routinely.

3.      Minor point: I assume the “Table 5” on page 9 is “Table 6”.

Author Response

Reviewer #1:

The manuscript titled” Immune-modified Glasgow prognostic score predicts therapeutic effect of pembrolizumab in recurrent and metastatic head and neck cancer” tried to prove the imGPS may be a used predictive and prognostic marker for the therapeutic effect of pembrolizumab for RMHNC. However, there are several defects leading the conclusion might be biased: 1. This is a retrospective study, thus, the power of imGPS on prediction will be limited. 2.  Although 54 patient samples are included, only a small number for each subtype. Besides, 81.5% of included patients are male.

→Thank you for the critical point. As you pointed out, this study has some biases and limitations, as described in the Discussion section (Lines 286-291). In the future, we plan to increase the number of cases and also consider prospective studies.

Other comments:

  1. Relative to mGPS, the imGPS added the lymphocyte into consideration. The NLR (Neutrophil-to-Lymphocyte Ratio) was less robust on prediction for prognosis of pembrolizumab than lymphocyte. I wonder if this is due to the variously changed neutrophil levels.

→We appreciate the important comment. As you pointed out, in this study, although lymphocyte count alone was able to predict prognosis, no significant differences were found in the NLR, which includes lymphocyte count. This may seem contradictory at first glance. HNCs develop close to the outside with frequent microorganisms contact, which induces inflammation. In addition, some advanced HNCs tend to show necrosis because of a lack of blood flow in the center region of the tumor. These factors influence neutrophil counts and may be related to the weak effectiveness of NLR compared to imGPS in correctly predicting pembrolizumab efficacy. We have added that sentence to Lines 257-263.These may be corrected in the future as the number of cases increases.

  1. It is great to know that the imGPS that could be employed with daily blood test. I wonder if the imGPS is dynamically changed after receiving treatment if monitored routinely.

→We are also very interested in how imGPS changes after treatment. There are some reports that post-treatment values ​​of some biomarkers are more useful for predicting prognosis (Reference No.29, 31), so it is expected that imGPS will also change dynamically after treatment. Unfortunately, some patients in this study didn’t check ALB in the follow-up blood tests, so we were unable to calculate imGPS after treatment.

  1. Minor point: I assume the “Table 5” on page 9 is “Table 6”.

→We have changed “Table 5” to “Table 6”.

Reviewer 2 Report

Comments and Suggestions for Authors

The authors improved the mGPS scale by adding a third component, the number of lymphocytes in the blood (imGPS scale), to the definition of CRP and albumin. Based on significant differences, including ROC analysis, the authors demonstrated the prognostic advantages of the imGPS scale in evaluating treatment with pembrolizumab for tumor disease - RMHNC. The results of this study are relevant to practical oncology. Meanwhile, I have a few comments: 

(1) It is customary in MDPI style to present links not in brackets, but in the form - [1,2].

(2) Subsection 2.1. Patients. Were the RMHNC treatment protocols with pembrolizumab standard for all patients, or did patients receive different antitumor therapy regimens?

(3) In Table 1, it is necessary to specify the units of measurement for each indicator in the table itself or in a note.

(4) Lines 87-90. "... and the number of positive cells (tumor cells, lymphocytes, and macrophages) / the total number of viable tumor cells". - and where is x 100? (DOI: 10.4103/crst.crst_306_22)

(5) Lines 106-107 To evaluate the usefulness of imGPS, ROC analysis was performed to calculate the area under the curve (AUC) value. What was the criterion of utility - the effect of pembrolizumab treatment sought? Overall survival (OS) and/or progression-free survival (PFS)?

(6) References must be in MDPI style.

Author Response

Reviewer #2:

The authors improved the mGPS scale by adding a third component, the number of lymphocytes in the blood (imGPS scale), to the definition of CRP and albumin. Based on significant differences, including ROC analysis, the authors demonstrated the prognostic advantages of the imGPS scale in evaluating treatment with pembrolizumab for tumor disease - RMHNC. The results of this study are relevant to practical oncology. Meanwhile, I have a few comments:

  • It is customary in MDPI style to present links not in brackets, but in the form - [1,2].

→ We have amended all parts of the manuscript to fit to MDPI style.

(2) Subsection 2.1. Patients. Were the RMHNC treatment protocols with pembrolizumab standard for all patients, or did patients receive different antitumor therapy regimens?

→We appreciate the important comment. Pembrolizumab is indicated for "head and neck cancer with recurrence or distant metastasis," so in this study, we treated patients according to the eligibile criteria. Basically, patients were divided into pembrolizumab-alone and chemotherapy-combination groups based on CPS, but the regimen was also selected while taking into consideration not only CPS but also age, Performance Status (PS), and tumor progression rate. In the chemotherapy combination group, cisplatin plus 5-fluorouracil or carboplatin plus 5-fluorouracil was selected based on renal function, and some cases included dose reduction. We have added that sentence to Lines 73-78.

(3) In Table 1, it is necessary to specify the units of measurement for each indicator in the table itself or in a note.

→Thank you for the comment. We have added the units of measurement for each indicator in Table 1.

(4) Lines 87-90. "... and the number of positive cells (tumor cells, lymphocytes, and macrophages) / the total number of viable tumor cells". - and where is x 100? (DOI: 10.4103/crst.crst_306_22)

→Thank you for the critical indication. We have added the sentence ”multiplied by 100” to Line 95.

(5) Lines 106-107 To evaluate the usefulness of imGPS, ROC analysis was performed to calculate the area under the curve (AUC) value. What was the criterion of utility - the effect of pembrolizumab treatment sought? Overall survival (OS) and/or progression-free survival (PFS)?

→In the ROC analysis, the usefulness criteria were prognosis (OS and PFS), which was added to Lines 112.

(6) References must be in MDPI style.

→ We have amended all parts of the manuscript to fit to MDPI style.

We appreciate the reviewer #1 and #2 for their insightful comments. After these revisions, we think the manuscript is greatly improved and more beneficial for readers.
